

# An integrable Lorentz-breaking deformation of two-dimensional CFTs

**Monica Guica**[1,2,3]

**1** Institut de Physique Théorique, CEA Saclay, 91191 Gif-sur-Yvette, France
**2** Department of Physics and Astronomy, Uppsala University,
SE-751 08 Uppsala, Sweden
**3** Nordita, Stockholm University and KTH Royal Institute of Technology,
Roslagstullsbacken 23, SE-106 91 Stockholm, Sweden

## Abstract

It has been recently shown that the deformation of an arbitrary two-dimensional conformal field theory by the composite irrelevant operator $T\bar{T}$, built from the components of the stress tensor, is solvable; in particular, the finite-size spectrum of the deformed theory can be obtained from that of the original CFT through a universal formula. We study a similarly universal, Lorentz-breaking deformation of two-dimensional CFTs that possess a conserved $U(1)$ current, $J$. The deformation takes the schematic form $J\bar{T}$ and is interesting because it preserves an $SL(2,\mathbb{R}) \times U(1)$ subgroup of the original global conformal symmetries. For the case of a purely (anti)chiral current, we find the finite-size spectrum of the deformed theory and study its thermodynamic properties. We test our predictions in a simple example involving deformed free fermions.

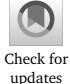

## 1. Introduction

Recently, Smirnov and Zamolodchikov [1] have studied an infinite class of irrelevant deformations of integrable two-dimensional QFTs (IQFTs), of the general double-trace form

$$S = S_{CFT} + \int_0^\mu d\mu' \int d^2z \, (T_s \bar{T}_{s'} - \Theta_{s-2}\bar{\Theta}_{s'-2})_{\mu'} \,, \qquad (1.1)$$

where $T_s, \Theta_{s-2}$ and their barred counterparts correspond to the components of conserved spin $s$ currents in the deformed QFT[1]. These deformations are integrable, in the sense that there still exists an infinite set of local integrals of motion. They are also very interesting because they represent an unusual type of flow "up the RG" direction.

The IQFT can also be taken to be a generic two-dimensional CFT, case in which the conserved currents and their associated commuting charges correspond to the KdV conserved integrals [2–4]. A particularly interesting deformation in this class is the $T\bar{T}$ deformation [1,5], built from the components of the stress tensor, which can be defined also for arbitrary two-dimensional QFTs and for which the finite-size spectrum and thermodynamic equation of state have been obtained exactly at finite $\mu$ in terms of the spectrum and, respectively, the equation of state of the original QFT.

$T\bar{T}$-deformed two-dimensional CFTs have a number of very interesting properties and applications. For positive $\mu$, the high-energy spectrum exhibits Hagedorn behaviour, which can be understood from the close relation between the $T\bar{T}$ deformation of free bosons and the worldsheet theory of the bosonic string [6,7]. For negative $\mu$, the theory appears to have a finite number of states and exhibits superluminal propagation [8,9]. The negative sign deformation has found a nice application in the AdS$_3$/CFT$_2$ correspondence, as a proposed holographic dual to $AdS_3$ gravity with a finite bulk cutoff, $r_c = 1/\sqrt{|\mu|}$ [9]. Another very interesting feature of $T\bar{T}$ - deformed QFTs is that, while their UV behaviour can be studied via their known S-matrix, it does not appear to correspond to a usual UV fixed point. Rather, it has been argued to correspond to a new type of UV behaviour, more characteristic of a theory of quantum gravity, termed "asymptotic fragility" [7,10]. A certain single-trace (in the AdS/CFT sense) variation on the $T\bar{T}$ deformation has also been argued to provide a holographic dual to string theory on a linear dilaton background [11]. For all of these reasons, the $T\bar{T}$ deformation is extremely interesting, in both field theory and holography.

It is natural to ask whether similarly integrable, universal double-trace deformations of two-dimensional CFTs exist. In this article, we focus on two-dimensional CFTs that posess an additional conserved $U(1)$ current, $J$, and consider the Lorentz-breaking double-trace deformation (2.1) constructed from $J$ and $\bar{T}$, where $\bar{T}$ is the current generating $\bar{z}$ translations. As we will show, we can again obtain an exact formula for the finite-volume spectrum of the deformed theory and work out the thermodynamics.

Apart from its solvability, this deformation is interesting because it preserves an $SL(2,\mathbb{R})_L \times U(1)_R$ subgroup[2] of the original global conformal group, as can be seen from the fact that the deforming operator has dimension $(1,2)$. Two-dimensional (local) QFTs with $SL(2,\mathbb{R})_L \times U(1)_R$ symmetry have been analysed in [12], where it was shown that, similarly to the case of two-dimensional CFTs [13], there is an infinite-dimensional enhancement of the global symmetry group. As one would expect, the left-moving conformal symmetry, $SL(2,\mathbb{R})_L$, is enhanced to a full left-moving Virasoro algebra; the surprising finding of [12] was that the right-moving translations $U(1)_R$ are *also* enhanced to an infinite symmetry group, which can be either a left-moving $U(1)$ Kač-Moody algebra or a right-moving Virasoro algebra, or both.

---

[1]Even though [1] only considered deformations by scalar operators ($s' = s$), the generalization of their results to deformations with nonzero total spin is straightforward.

[2]Strictly speaking, the $U(1)_R$ is non-compact, but we keep this notation to connect with earlier literature.

Two-dimensional QFTs with the former symmetry enhancement pattern are known as "warped CFTs" [14, 15]. They are invariant under the change of coordinates

$$z \to f(z), \quad \bar{z} \to \bar{z} + g(z),  \tag{1.2}$$

where $f(z), g(z)$ are two arbitrary holomorphic functions. Their properties have been studied in [14–18], and they were in particular shown to exhibit an interesting Cardy-like growth of their density of states.

However, the theory defined via (2.1), to the extent that it can be understood as a quasilocal two-dimensional QFT (e.g., below the cutoff scale $\mu$) such that the results of [12] apply, is rather expected to fall in the second category, where the $\bar{z}$ translations are enhanced to an infinite-dimensional right-moving Virasoro symmetry. The reason for this expectation is that the deformation parameter $\mu$ is continuous, and it would be quite surprising if the right-moving Virasoro symmetry of the original CFT suddenly became a left-moving Kač-Moody symmetry as we infinitesimally turn on $\mu$. This expectation can in principle be confirmed via holographic calculations [19] or directly, e.g. by using conformal perturbation theory. If the expectation is confirmed, then the theory defined via (2.1) would represent the first non-trivial example of a QFT with only $SL(2, \mathbb{R})_L \times U(1)_R$ global symmetry that is enhanced to Virasoro$_L$ × Virasoro$_R$. However, in this article we will limit ourselves to defining the deformed theory, working out its spectrum and thermodynamics, and we leave the interesting question of symmetry enhancement to later work.

The plan of this paper is as follows. In section 2., we discuss the basics of the $J\bar{T}$ deformation and work out the spectrum and basic thermodynamic properties of the deformed theory in finite volume, after specializing to a purely (anti)chiral $U(1)$ current. In section 3., we check our general prediction for the deformed spectrum in a concrete example involving deformed free fermions. Section 4. contains a discussion and future directions. In the appendix, we study an $SL(2, \mathbb{R})_L \times U(1)_R$ invariant deformation of the classical free boson and underline some of its interesting features.

## 2. The $J\bar{T}$ deformation

Consider a two-dimensional CFT with a $U(1)$ symmetry generated by a conserved current $J$. We consider the following double-trace irrelevant deformation

$$S = S_{CFT} + \int_0^\mu d\mu' \int d^2z \, (\mathcal{O}_{J\bar{T}})_{\mu'}, \qquad \mathcal{O}_{J\bar{T}} = J\bar{T} - \bar{J}\Theta,  \tag{2.1}$$

where $J = J_z$ and $\bar{J} = J_{\bar{z}}$ are the components of the $U(1)$ current and $\bar{T} = T_{\bar{z}\bar{z}}$ and $\Theta = T_{z\bar{z}}$ are the components of the current generating $\bar{z}$ translations in the deformed QFT. We assume, as in [1], that the deformed theory can be understood as a (quasi)local two-dimensional QFT below some scale, so the local currents $(J, \bar{J})$ and $(\bar{T}, \Theta)$ continue to exist. The currents satisfy the conservation equations

$$\partial \bar{T} + \bar{\partial}\Theta = 0, \quad \partial \bar{J} + \bar{\partial} J = 0,  \tag{2.2}$$

The double-trace operator $\mathcal{O}_{J\bar{T}}$ is defined via the OPE[3]

$$J(z)\bar{T}(z') - \bar{J}(z)\Theta(z') \sim \mathcal{O}_{J\bar{T}}(z') + \text{total derivative terms} . \qquad (2.3)$$

This form of the OPE follows from the fact that both the $z$ and $z'$ derivatives of (2.3) can be shown, using the conservation equations (2.2), to only contain total derivative terms [21]. This allows for only one operator that is not a total derivative itself to appear in the OPE, and it must have a constant coefficient.

The theory also has a current $T^\lambda{}_z$ associated with $z$ translations, satisfying

$$\partial_z T_{\bar{z}z} + \partial_{\bar{z}} T_{zz} = 0 . \qquad (2.4)$$

If the deformation preserves full $SL(2,\mathbb{R})_L$ invariance, then the current $j_L^\lambda = T^\lambda{}_z \cdot z$ is also conserved, implying that $T_{\bar{z}z} = 0$, which is equivalent with holomorphy of $T_{zz}$. The latter can in principle be checked using conformal perturbation theory. At linear order in $\mu$, $SL(2,\mathbb{R})_L$ invariance is obviously preserved because the perturbing operator has $h = 1$. It seems reasonable that $J\bar{T}$ will stay exactly marginal on the left at higher orders in the deformation parameter, though it would be interesting to prove this statement, using techniques such as those of [22, 23].

A holomorphic $T = T_{zz}$ implies the existence of an infinite-dimensional left-moving Virasoro symmetry enhancing the $SL(2,\mathbb{R})_L$. Additionally, if the internal $U(1)$ current is purely chiral ($\bar{J} = 0$) or purely antichiral ($J = 0$), then we also expect an infinite enhancement of the $U(1)$ symmetry to either a left or a right-moving $U(1)$ Kač-Moody symmetry.

## 2.1 The finite-size spectrum

We now place the deformed theory on a cylinder of circumference $R$ and study its spectrum, which in general will be discrete. We denote the cylinder coordinates as $t, \varphi$, with $\varphi \sim \varphi + R$. We will mostly work in Euclidean signature (assuming we can Wick rotate), with Euclidean time $\tau = -it$. The holomorphic coordinate $z$ is given by[4]

$$z = \varphi + i\tau , \quad z \sim z + R . \qquad (2.5)$$

Following [1], we consider eigenstates of the Hamiltonian $H$, momentum operator $P$ and charge operator $Q$

$$H = \int_0^R d\varphi\, T_{tt} , \qquad P = \int_0^R d\varphi\, T_{t\varphi} , \qquad Q = \int_0^R d\varphi\, J_t , \qquad (2.6)$$

which commute and thus can be simultaneously diagonalized. We denote these eigenstates collectively as $|n\rangle$

$$H|n\rangle = E_n|n\rangle , \quad P|n\rangle = P_n|n\rangle , \quad Q|n\rangle = Q_n|n\rangle . \qquad (2.7)$$

---

[3]Note that, strictly speaking, this OPE argument is valid when the deformed CFT is local in the UV, which is not the case for either $T\bar{T}$ or $J\bar{T}$. In particular, at scales comparable to $\mu$ it is not even clear how to define the currents $J$, $\bar{T}$. The derivation we present is thus only justified at scales larger than $\mu$ and it provides a simple operatorial way to derive the deformed spectrum, which in the $T\bar{T}$ case has been checked against a variety of other methods [7, 20].

[4]The holomorphic coordinate on the cylinder is related to the non-compact coordinate on the plane via the usual map $z_{pl} = \exp(-2\pi i z_{cyl}/R)$, which is a symmetry of the theory. The map between the plane and the cylinder for $SL(2,\mathbb{R}) \times U(1)$ invariant two-dimensional QFTs has been previously discussed in the context of warped CFTs in [16]. In that case, the second symmetry in (1.2) ensures that the theories on the plane and on the cylinder are equivalent; however, in our case this symmetry is absent, and it would be interesting to better understand the relationship between the two.

The expectation value of the deforming operator in the above eigenstate can be computed from the correlator

$$\mathcal{C}_n(z, z') = \langle n|J(z)\bar{T}(z') - \bar{J}(z)\Theta(z')|n\rangle \,, \tag{2.8}$$

which can be shown to be independent of $z, z'$. In the limit $z \to z'$, this correlator simply reduces to the expectation value of the deforming operator in the state $|n\rangle$, $\langle n|\mathcal{O}_{J\bar{T}}|n\rangle$. One can also evaluate $\mathcal{C}_n(z, z')$ by inserting a complete set of energy-momentum-charge eigenstates $|n'\rangle\langle n'|$ in between the two operators

$$\mathcal{C}_n(z, z') = \sum_{n'} \langle n|J(z)|n'\rangle\langle n'|\bar{T}(z')|n\rangle - \langle n|\bar{J}(z)|n'\rangle\langle n'|\Theta(z')|n\rangle \,. \tag{2.9}$$

Expanding around $z$, one can show that $\mathcal{C}_n(z, z')$ can only be $z'$-independent if all contributions with $n' \neq n$ cancel out [21]. Assuming the spectrum is non-degenerate[5], we thus have

$$\langle n|\mathcal{O}_{J\bar{T}}|n\rangle = \langle n|J|n\rangle\langle n|\bar{T}|n\rangle - \langle n|\bar{J}|n\rangle\langle n|\Theta|n\rangle \,. \tag{2.10}$$

It is useful to write this relation in terms of the components of the stress tensor along the Euclidean coordinates $\varphi, \tau$. We have

$$T_{zz} = \frac{1}{4}[T_{\varphi\varphi} - T_{\tau\tau} - i(T_{\tau\varphi} + T_{\varphi\tau})] \,, \qquad T_{\bar{z}z} = \frac{1}{4}[T_{\varphi\varphi} + T_{\tau\tau} + i(T_{\tau\varphi} - T_{\varphi\tau})] \,, \tag{2.11}$$

$$T_{\bar{z}\bar{z}} = \frac{1}{4}[T_{\varphi\varphi} - T_{\tau\tau} + i(T_{\tau\varphi} + T_{\varphi\tau})] \,, \qquad T_{z\bar{z}} = \frac{1}{4}[T_{\varphi\varphi} + T_{\tau\tau} + i(T_{\varphi\tau} - T_{\tau\varphi})] \,. \tag{2.12}$$

Note that since the theory is not Lorentz invariant, in general $T_{\varphi\tau} \neq T_{\tau\varphi}$. However, due to the $SL(2, \mathbb{R})_L$ scaling symmetry we have argued for, we have instead $T_{\bar{z}z} = 0$. This allows us to solve for $T_{\varphi\tau}$ in terms of the other components, so the above equations become

$$T_{zz} = -\frac{1}{2}(T_{\tau\tau} + iT_{\tau\varphi}) \,, \qquad T_{\bar{z}\bar{z}} = \frac{1}{2}(T_{\varphi\varphi} + iT_{\tau\varphi}) \,, \qquad T_{z\bar{z}} = \frac{1}{2}(T_{\varphi\varphi} + T_{\tau\tau}) \,. \tag{2.13}$$

The current components read

$$J = \frac{1}{2}(J_\varphi - iJ_\tau) \,, \qquad \bar{J} = \frac{1}{2}(J_\varphi + iJ_\tau) \,, \tag{2.14}$$

and the expectation value of $\mathcal{O}_{J\bar{T}}$ can be written as

$$\langle n|\mathcal{O}_{J\bar{T}}|n\rangle = \frac{1}{4}\langle n|(J_\varphi - iJ_\tau)|n\rangle\langle n|T_{\varphi\varphi} + iT_{\tau\varphi}|n\rangle - \frac{1}{4}\langle n|(J_\varphi + iJ_\tau)|n\rangle\langle n|T_{\varphi\varphi} + T_{\tau\tau}|n\rangle \,. \tag{2.15}$$

As $\mu$ is infinitesimally changed, the deformation (2.1) of the euclidean action induces a change in the energy levels, $E_n$, of the form

$$\frac{\partial E_n(\mu, R)}{\partial \mu} = 2\int_0^R d\varphi \, \langle n|\mathcal{O}_{J\bar{T}}|n\rangle = 2R \, \langle n|\mathcal{O}_{J\bar{T}}|n\rangle \,, \tag{2.16}$$

which can be derived from the change with $\mu$ of the partition function. The factor of 2 is due to the change of measure, $d^2z = 2\, d\tau d\varphi$. As in [1, 5], (2.16) will be the essential equation allowing us to compute the exact finite-size spectrum of the deformed theory. The expectation

---

[5]In many theories of interest, the states of charge $Q$ and $-Q$ will have the same energy, so we may worry about the terms in the sum with $|n'\rangle = |E, -Q\rangle$. However, using the fact that $[Q, T_{\alpha\beta}] = 0$, it is easy to show that the expectation value of $\langle Q, E|T_{\alpha\beta}|-Q, E\rangle = 0$, so these terms will drop out from the sum.

values of the current components in the translationally-invariant state $|n\rangle$ are related to the corresponding conserved charges as

$$\langle n|T_{\tau\tau}|n\rangle = -\frac{E_n}{R}\,, \qquad \langle n|T_{\varphi\varphi}|n\rangle = -\frac{\partial E_n}{\partial R}\,, \qquad \langle n|T_{\tau\varphi}|n\rangle = \frac{iP_n}{R}\,, \qquad \langle n|J_{\tau}|n\rangle = \frac{iQ_n}{R}\,. \quad (2.17)$$

Note that, in order to determine the spectrum, we still need an equation for $\langle n|J_{\varphi}|n\rangle$ in terms of conserved quantities. However, unlike for the above, there does not appear to exist a general formula relating $\langle n|J_{\varphi}|n\rangle$ to only the conserved charges. Therefore, we will determine the spectrum by making the additional simplifying assumption that the current is purely (anti)holomorphic. We treat each case separately below.

**Purely holomorphic current**

Assuming that the current is purely holomorphic, which implies $J_{\varphi} = -iJ_{\tau}$, we find

$$\langle n|\mathcal{O}_{J\bar{T}}|n\rangle = -\frac{Q_n}{2R}\left(\frac{\partial E_n}{\partial R} + \frac{P_n}{R}\right)\,. \quad (2.18)$$

Thus, when changing $\mu$ infinitesimally, the energy levels change as

$$\frac{\partial E_n(\mu,R)}{\partial \mu} = -Q_n\left(\frac{\partial E_n}{\partial R} + \frac{P_n}{R}\right)\,, \quad (2.19)$$

which is the equation that we need to solve. Note that since $P_n$ is quantized, $P_nR \in \mathbb{Z}$, it cannot vary with $\mu$. It is useful to introduce the left/right-moving energies $E_n^{L,R}$ via

$$E_n = E_n^L + E_n^R\,, \qquad P_n = E_n^L - E_n^R\,. \quad (2.20)$$

In terms of $E_n^R$, the level equation is

$$\frac{\partial E_n^R}{\partial \mu} + Q_n\frac{\partial E_n^R}{\partial R} = 0\,. \quad (2.21)$$

Assuming that $Q_n$ is quantized and thus $\mu$-independent, the general solution is

$$E_n^R(\mu,R) = E_n^R(0,R-\mu Q_n)\,. \quad (2.22)$$

The left-moving energy is simply given by

$$E_n^L = E_n^R + P_n\,. \quad (2.23)$$

In the original undeformed CFT, the left/right energies are given by

$$E^R(0,R) = 2\pi \cdot \frac{\bar{h} - \frac{c}{24}}{R}\,, \quad (2.24)$$

$$E^L(0,R) = 2\pi \cdot \frac{h - \frac{c}{24}}{R} = E^R(0,R) + 2\pi \cdot \frac{h - \bar{h}}{R}\,, \quad (2.25)$$

where $h, \bar{h}$ (and $Q$) label the undeformed CFT spectrum with $h - \bar{h} \in \mathbb{Z}/2$ and we dropped the index '$n$'. Consequently, in the deformed theory the left/right energies will be

$$E^R(\mu,R) = 2\pi \cdot \frac{\bar{h} - \frac{c}{24}}{R - \mu Q}\,, \quad (2.26)$$

$$E^L(\mu, R) = E^R(\mu, R) + 2\pi \cdot \frac{h - \bar{h}}{R} = 2\pi \cdot \frac{h - \frac{c}{24}}{R} + 2\pi \cdot \frac{\mu Q\left(\bar{h} - \frac{c}{24}\right)}{R(R - \mu Q)} \; . \tag{2.27}$$

Thus, the energies of all states that carry a non-trivial left-moving charge $Q$ are modified: they grow for $\mu Q > 0$ and decrease for $\mu Q < 0$. States with $Q = 0$ are undeformed; in particular, all the left-moving ground states, which have $Q = h = 0$, are unchanged. Since in a generic CFT containing a $U(1)$ current the lowest-lying charged state must have $h < c/\alpha + \mathcal{O}(1)$ with $\alpha > 8$ [24], the spectrum will necessarily be modified above this conformal dimension.

The effective local description breaks down for $R < \mu Q$, or equivalently $Q > R/\mu$. Under the natural assumption that the CFT spectrum is symmetric under $Q \to -Q$, we find that the deformed finite-size theory breaks down at this radius for either sign of $\mu$. This is quite different from the $T\bar{T}$ case, where one obtains radically different behaviour for each of the signs of $\mu$. Also note that, in assuming that the charge $Q$ is quantized, we have ignored the effect of possible chiral anomalies, which can have an important effect on the spectrum [25].

**Purely antiholomorphic current**

If we now assume that the current is purely antiholomorphic ($J_\varphi = +iJ_\tau$), we find

$$\langle n|\mathcal{O}_{JT}|n\rangle = -\frac{Q_n}{2R}\left(\frac{\partial E_n}{\partial R} + \frac{E_n}{R}\right) , \tag{2.28}$$

which yields the following equation for the energy levels

$$\frac{\partial E_n}{\partial \mu} = -Q_n\left(\frac{\partial E_n}{\partial R} + \frac{E_n}{R}\right) . \tag{2.29}$$

Letting $\varepsilon_n(\mu, R) \equiv E_n(\mu, R)R$, we find

$$\frac{\partial \varepsilon_n}{\partial \mu} = -Q\frac{\partial \varepsilon_n}{\partial R} \quad \Rightarrow \quad \varepsilon_n(\mu, R) = \varepsilon_n(R - \mu Q) . \tag{2.30}$$

However, since in a CFT $\varepsilon_n$ is independent of $R$, we find that in this case the spectrum is entirely undeformed. This can be understood from the fact that in the original CFT, the operator $T_{z\bar{z}}$ that enters the deformation is zero inside correlation functions, so all corrections to the partition function in conformal perturbation theory vanish.

## 2.2 Thermodynamics

Let us start with the case of a purely holomorphic current, where the energy levels are non-trivially displaced, as in (2.27). Since the spectrum is continuously deformed, we expect that the degeneracy is still given by the CFT formula[6]

$$S = 2\pi\sqrt{\frac{c}{6}\left(h - \frac{c}{24} - \frac{Q^2}{k}\right)} + 2\pi\sqrt{\frac{c}{6}\left(\bar{h} - \frac{c}{24}\right)}, \tag{2.31}$$

which holds for $h, \bar{h} >> c$. Here $k$ is the level of the left Kač-Moody chiral algebra, and we are working in the limit $k \to 0$. Setting for simplicity the total momentum to zero, we have

$$E = E_L + E_R = 2\pi \cdot \frac{2(h - \frac{c}{24})}{R - \mu Q} \; . \tag{2.32}$$

---

[6]This is because the levels of fixed $h, \bar{h}$ and $Q$ do not cross as we vary $\mu$. Note that if $Q$ were not fixed, then levels with $h' > h$ may cross if $Q' < Q$.

Therefore, in terms of $E, Q$, the expression for the entropy is

$$S = 2\pi\sqrt{\frac{c}{12}\left(E(R - \mu Q) - \frac{2Q^2}{k}\right)} + 2\pi\sqrt{\frac{c}{12}E(R - \mu Q)} \equiv 2\pi(S_L + S_R) \,. \tag{2.33}$$

The first law of thermodynamics reads

$$T dS = dE + p dR - \Phi dQ \,. \tag{2.34}$$

Thus, the temperature is given by

$$T = \left(\frac{\partial S}{\partial E}\right)_{R,Q}^{-1} = \frac{12}{\pi c(R - \mu Q)}\frac{S_L S_R}{S_L + S_R} \,. \tag{2.35}$$

Notice this blows up as $R$ approaches $\mu Q$. The "pressure" is

$$p = T\left(\frac{\partial S}{\partial R}\right)_{Q,E} = \frac{\pi c E\, T}{12}\frac{S_L + S_R}{S_L S_R} = \frac{E}{R - \mu Q} = -\frac{\partial E}{\partial R} \,, \tag{2.36}$$

as expected. Finally, the chemical potential

$$\Phi = -T\left(\frac{\partial S}{\partial Q}\right)_{R,E} = \frac{\mu E + 4QS_R/(k(S_L + S_R))}{R - \mu Q} \,. \tag{2.37}$$

Thus, we see that all the thermodynamic quantities diverge at $R = \mu Q$, as the gap between energy levels becomes infinite at small enough but finite radius. Effectively, it looks like the theory lives on a circle of radius $R - \mu Q$, rather than $R$.

This observation can be formalized by noticing that, at least at a perturbative level, the $J\bar{T}$ deformation can be induced via a field-dependent diffeomorphism performed on the original two-dimensional CFT[8]

$$z \to z' = z \,, \qquad \bar{z} \to \bar{z}' = \bar{z} - \mu\int^z J(w)\,dw \,. \tag{2.38}$$

Since the coordinates of the deformed theory are identified as $z \sim z + R$, $\bar{z} \sim \bar{z} + R$, the identifications in the undeformed picture are

$$z' \sim z' + R \,, \qquad \bar{z}' \sim \bar{z}' + R - \mu Q \,, \tag{2.39}$$

which tells us that the right-movers experience a different size of the circle. We can also understand this as a change of the metric on which the CFT is placed [26]

$$ds^2 = dz\,d\bar{z}' = dz\,(d\bar{z} - \mu J(z)\,dz) = d\varphi^2 - dt^2 - \mu J(\varphi + t)(d\varphi + dt)^2 \,. \tag{2.40}$$

Assuming that the current is constant, $J = Q/R$, we find that the metric develops closed time-like curves for $R < \mu Q$, which is another way to see the breakdown of the theory at this radius.

---

[7] Note this transformation would be a symmetry of warped CFTs, though it is not a symmetry here.

[8] The variation of the action under the coordinate transformation $x^a \to x^a + \xi^a(x)$ is given by

$$\delta S = -\int d^d x\, T^\lambda{}_a\,\partial_\lambda \xi^a + total\ deriv.$$

which agrees with the expression used in the following sections, but differs from the usual definition via the coupling to a background vielbein, $\delta S = \int d^d x\, e\, T_\mu{}^a\,\delta e_a^\mu$, by a factor of $e = \sqrt{g}$. (Since the stress tensor is not symmetric, it most naturally couples to a background vielbein).

The deformation also leads to a modification of the propagation speed of excitations, which can now become superluminal around certain backgrounds. The speed of the left/right-moving excitations in the metric (2.40) is

$$c_s^L = 1 \,, \quad c_s^R = \frac{R + \mu Q}{R - \mu Q} \approx 1 + \frac{2\mu Q}{R} + \mathcal{O}(\mu^2) \,. \tag{2.41}$$

Notice that the propagation speed for the right-movers is superluminal for $\mu Q > 0$. This can be understood from the fact that the interaction is repulsive in this range[9] [8, 26]. Note that superluminal propagation does not in itself pose a problem, because the deformed theory is not Lorentz invariant. However, it does lead to problems in the finite volume theory, as (2.40) shows that points in the same constant time slice cannot be identified if $Q > R/\mu$.

For the case of a purely right-moving current, we have at zero momentum

$$S = 2\pi \sqrt{\frac{cER}{12}} + 2\pi \sqrt{\frac{c}{12}\left(ER - \frac{2Q^2}{k}\right)} \,, \tag{2.42}$$

which is identical to the entropy in a CFT with a right-moving Kač-Moody current of level $k$. Notice that this deformation can be induced by the field-dependent coordinate transformation

$$z \to z \,, \quad \bar{z} \to \bar{z} + \mu \int^{\bar{z}} d\bar{z}' \bar{J}(\bar{z}') \,, \tag{2.43}$$

which is entirely antiholomorphic. Under it, the right-moving stress tensor picks up a factor proportional to $2\mu c \bar{J}''(\bar{z})$. Since the latter integrates to zero, the energy levels are unchanged.

Thus, we find that the spectrum and thermodynamics of the deformed theory depend on the properties of the current by which we deform, e.g. chiral vs. anti-chiral. Below, we study some simple examples.

## 3. A simple example: deformed free fermions

In this section, we would like to exemplify and check our general findings from the previous section. The simplest examples of theories where the $U(1)$ current can be made purely chiral/antichiral are fermionic ones. We treat the case of a purely chiral (left-moving) and purely antichiral (right-moving) $U(1)$ current separately.

### 3.1 Purely left-moving current

We consider the following action describing two complex fermions

$$S = \frac{i}{2} \int dt\, d\varphi \left( \bar{\partial}\psi_L \psi_L^\star - \psi_L \bar{\partial}\psi_L^\star + \psi_R \partial\psi_R^\star - \partial\psi_R \psi_R^\star + \mu\, \psi_L \psi_L^\star \left( \psi_R \bar{\partial}\psi_R^\star - \bar{\partial}\psi_R \psi_R^\star \right) \right). \tag{3.1}$$

---

[9]Possibly the simplest way [8] to verify that the interaction is repulsive is by computing the "binding energy" of a two-particle state in the deformed theory, as compared to the undeformed one (a free boson is a good example). The binding energy is given by $E_{bind} = E(2, \mu) - E(0, \mu) - 2\big(E(1, \mu) - E(0, \mu)\big)$. Plugging in (2.24) for $E(n, \mu)$ with $h = n$, we find that this quantity is positive for $\mu Q > 0$, so the interaction is repulsive.

At $\mu = 0$, this simply describes a free left-moving complex femion $\psi_L$ and a free right-moving complex fermion $\psi_R$. The purely holomorphic conserved current that we will be considering for the deformation is associated to the symmetry that rotates the left-moving fermions $\psi_L \rightarrow e^{i\alpha}\psi_L$, $\psi_L^\star \rightarrow e^{-i\alpha}\psi_L^\star$ with the right-moving ones $\psi_R, \psi_R^\star$ inert

$$J_z = \psi_L \psi_L^\star \equiv J_L \, . \tag{3.2}$$

The components of the stress tensor are[10]

$$T_{zz} = \frac{i}{4}\left(\partial\psi_L\psi_L^\star - \psi_L\partial\psi_L^\star\right) + \frac{i\mu}{4}\psi_L\psi_L^\star(\psi_R\partial\psi_R^\star - \partial\psi_R\psi_R^\star) \, , \tag{3.3}$$

$$T_{\bar{z}z} = -\frac{i}{4}\left(\bar{\partial}\psi_L\psi_L^\star - \psi_L\bar{\partial}\psi_L^\star + \mu\,\psi_L\psi_L^\star(\psi_R\bar{\partial}\psi_R^\star - \bar{\partial}\psi_R\psi_R^\star)\right) \, , \tag{3.4}$$

$$T_{z\bar{z}} = -\frac{i}{4}(\psi_R\partial\psi_R^\star - \partial\psi_R\psi_R^\star) \, , \tag{3.5}$$

$$T_{\bar{z}\bar{z}} = \frac{i}{4}(\psi_R\bar{\partial}\psi_R^\star - \bar{\partial}\psi_R\psi_R^\star) \equiv T_R \, . \tag{3.6}$$

Note that since the current is exactly holomorphic, the perturbation of the free fermion action takes the form of precisely $J\bar{T}$, after taking into account the change in the measure.

The equations of motion are

$$\bar{\partial}\psi_L = -\frac{\mu}{2}\psi_L(\psi_R\bar{\partial}\psi_R^\star - \bar{\partial}\psi_R\psi_R^\star) = 2i\mu\psi_L T_R \, , \tag{3.7}$$

$$\partial\psi_R + \mu\left(J_L\bar{\partial}\psi_R + \frac{1}{2}\bar{\partial}J_L\,\psi_R\right) = 0 \, , \tag{3.8}$$

and their starred counterparts. Note that on-shell we have $T_{\bar{z}z} = 0$, as expected. Also, we find that $T_{z\bar{z}} = \mu J_L T_{\bar{z}\bar{z}}$.

We start by solving for the currents $J_L, T_R$, which satisfy

$$\bar{\partial}J_L = 0 \, , \quad \partial T_R + \mu J_L\bar{\partial}T_R = 0 \, , \tag{3.9}$$

with the general solution

$$J_L = J_L(z) \, , \quad T_R = T_R\left(\bar{z} - \mu\int^z J_L(z')\,dz'\right) \, . \tag{3.10}$$

The solution for the fermions themselves is

$$\psi_R(z,\bar{z}) = \psi_R\left(\bar{z} - \mu\int^z J_L(z')\,dz'\right) \, , \tag{3.11}$$

and

$$\psi_L(z,\bar{z}) = e^{2i\mu S(z,\bar{z})}\psi_L^{(0)}(z) \, , \quad \bar{\partial}S = T_R \, . \tag{3.12}$$

To find the spectrum, we expand the fermions in modes, upon imposing appropriate boundary conditions. These will be either Ramond or Neveu-Schwarz

$$\psi_{L,R}(\varphi + R) = \pm\psi_{L,R}(\varphi) \, . \tag{3.13}$$

---

[10]The stress tensor is given by the usual formula, $T^\lambda{}_a = \frac{\partial\mathcal{L}}{\partial(\partial_\lambda\phi^i)}\partial_a\phi^i - \delta^\lambda_a\mathcal{L}$. Even though we are using the (anti)holomorphic notation, we are working in Lorentzian signature with $z \rightarrow x^+ = \varphi + t$ and $\bar{z} \rightarrow x^- = \varphi - t$.

The mode expansion for $\psi_R$ takes the form

$$\psi_R\left(\bar{z} - \mu \int^z dz' J_L(z')\right) = \sum_n \gamma_n b_n \exp\left(2\pi i n \frac{\bar{z} - \mu \int^z dz' J_L(z')}{R - \mu Q}\right), \qquad (3.14)$$

where $b_n$ represent fermionic creation/annihilation operators, satisfying

$$\{b_m, b_n^\dagger\} = \delta_{m,n}, \qquad (3.15)$$

and the constants $\gamma_n$ are normalization factors that will be determined shortly. The sum runs over $n$ integer in the Ramond sector, and over $n$ integer plus a half in the NS one. The shift in the radius in the denominator comes from the fact that under $\varphi \to \varphi + R$, the argument of $\psi_R$ shifts by $R - \mu Q$, where we are considering states with fixed $\psi_L^{(0)}$ charge $Q$.

The normalization factors $\gamma_n$ are determined from the equal-time commutation relations

$$\{\psi_R(\varphi), \pi_R(\varphi')\} = i\,\delta(\varphi - \varphi'), \qquad (3.16)$$

where the momentum canonically conjugate to $\psi_R$ is

$$\pi_R = \frac{i}{2}(1 - \mu J_L)\psi_R^\star, \qquad (3.17)$$

and similarly for its complex conjugate. Since the operator $J_L$ commutes with $\psi_R^\star$, there is no ordering ambiguity. We find[11]

$$|\gamma_m| = \sqrt{\frac{2}{R - \mu Q}}. \qquad (3.18)$$

Finally, we can now check whether the energy spectrum agrees with what we have derived on general grounds. The right-moving energy is given by

$$E^R \equiv \frac{1}{2}(E - P) = \int_0^R d\varphi\,(T_{\bar{z}\bar{z}} - T_{z\bar{z}}) = \int_0^R d\varphi\,T_R(1 - \mu J_L). \qquad (3.19)$$

Note that as far as the non-zero modes of $T_R$ are concerned, the integrand equals $\partial_\varphi S$, where $S$ has been defined in (3.12). Consequently, the energy only gets contributions from the zero modes, and reads

$$E^R = R\,T_{z.m.}^R\left(1 - \frac{\mu Q}{R}\right). \qquad (3.20)$$

Plugging in the mode expansion (3.14), the final expression for the right-moving energy takes the form

$$E^R = \pi \sum_m m\,|\gamma_m|^2 \langle b_m b_m^\dagger\rangle = \frac{E^R(0, R) \cdot R}{R - \mu Q}, \qquad (3.21)$$

where the expectation values is computed in an energy eigenstate, obtained by acting with a number of fermionic creation operators on the vacuum. We thus find a nice match with the general prediction (2.24).

---

[11]We used the identity

$$\delta(f(\varphi) - f(\varphi')) = \frac{1}{|f'(\varphi')|}\,\delta(\varphi - \varphi') = \frac{1}{R_f}\sum_{m=-\infty}^{\infty} e^{2\pi i m(f(\varphi) - f(\varphi'))/R_f}$$

with $f(\varphi) = \bar{z} - \mu \int^z dz' J_L(z')\big|_{z=\bar{z}=\varphi}$, $R_f = f(\varphi + R) - f(\varphi)$, as well as (3.15).

We can also match the prediction (2.27) for the left-moving energy

$$E^L \equiv \frac{1}{2}(E+P) = \int_0^R d\varphi \, T_{zz} \, . \tag{3.22}$$

Plugging in the solution (3.12) $\psi_L$ into $T_{zz}$, we find

$$T_{zz} = T_L^{(0)}(z) - \mu J_L \partial S - \mu^2 J_L^2 T_R \, , \tag{3.23}$$

with

$$T_L^{(0)}(z) = \frac{i}{4}\left(\partial \psi_L^{(0)} \psi_L^{\star(0)} - \psi_L^{(0)} \partial \psi_L^{\star(0)}\right) \, . \tag{3.24}$$

It is easy to check it satisfies $\bar{\partial} T_{zz} = 0$. Requiring (anti)-periodicity of the left-moving fermion solution (3.12), we find that $\psi_L^{(0)}(z)$ must have a mode expansion of the form $\exp(2\pi i n z/R)$. The part that depends on $T_L^{(0)}$ then gives a contribution $E^L(0,R)$ identical to the free fermion. The $\mu$-dependent correction to the energy is

$$\Delta E^L = -\mu \int d\varphi \left\langle J_L \left(\partial S + \mu J_L \bar{\partial} S\right)\right\rangle \, , \tag{3.25}$$

which only receives contributions from the zero modes of $S$. Requiring the absence of winding modes fixes $S_{z.m.} = T_{z.m.}^R(\bar{z} - z)$. Since it involves (free) fermions of two different types, the correlator factorizes and we find

$$E_L = E_L(0,R) + \mu Q \langle T_R \rangle \left(1 - \frac{\mu Q}{R}\right) = E^L(0,R) + \mu Q \frac{E^R(0,R)}{R - \mu Q} \, , \tag{3.26}$$

which agrees with (2.27), obtained via the general analysis. Note that in our manipulations above we have been rather cavalier about normal-ordering issues, which yield corrections proportional to the coefficient of the chiral anomaly. Thus, the match we found between the general $J\bar{T}$-deformed CFT spectrum and deformed chiral fermion spectra is contingent upon having consistently ignored the chiral anomaly in both analyses.

## 3.2 Purely right-moving current

We now consider the model

$$S = i \int dt d\varphi \left[\psi_1 \partial \psi_1 \left(1 + \mu \psi_2 \psi_2^\star\right) + \frac{1}{2}(\psi_2 \partial \psi_2^\star - \partial \psi_2 \psi_2^\star)\right] \, , \tag{3.27}$$

where $\psi_1$ is a real two-dimensional fermion, $\psi_2$ is a complex fermion and $J$ is the current associated to the symmetry $\psi_2 \to e^{i\alpha}\psi_2$, $\psi_2^\star \to e^{-i\alpha}\psi_2^\star$

$$J_{\bar{z}} = \psi_2 \psi_2^\star \equiv \bar{J} \, , \tag{3.28}$$

which is purely antiholomorphic. The components of the stress tensor read

$$T_{zz} = T_{\bar{z}z} = 0 \, , \quad T_{z\bar{z}} = -\frac{i}{2}\left[\psi_1 \partial \psi_1 \left(1 + \mu \psi_2 \psi_2^\star\right) + \frac{1}{2}(\psi_2 \partial \psi_2^\star - \partial \psi_2 \psi_2^\star)\right] \, , \tag{3.29}$$

$$T_{\bar{z}\bar{z}} = \frac{i}{2}\left[\psi_1 \bar{\partial} \psi_1 \left(1 + \mu \psi_2 \psi_2^\star\right) + \frac{1}{2}(\psi_2 \bar{\partial} \psi_2^\star - \bar{\partial} \psi_2 \psi_2^\star)\right] \, . \tag{3.30}$$

The equations of motion imply that $\psi_{1,2}$ are anti-holomorphic, so $T_{z\bar{z}} = 0$ on-shell. It is not hard to check that the deformation takes the form of a $J\bar{T}$-type deformation, since the Lagrangian satisfies [5]

$$\partial_\mu \mathcal{L} = \mathcal{O}_{J\bar{T}} = -J_{\bar{z}} T_{z\bar{z}} \tag{3.31}$$

upon taking into account the Grassman nature of $\psi_2$.

The action (3.27) differs from the free fermionic action for $\psi_{1,2}$ by the purely antiholomorphic coordinate transformation[12]

$$\bar{z} \to \bar{z}' = \bar{z} + \mu \int^{\bar{z}} d\bar{w} \, \bar{J}(\bar{w}) \,. \tag{3.32}$$

Note that $\bar{z}' \sim \bar{z}' + R'$, where $R' = R - \mu Q$ in the superselection sector of charge $Q = -\int_0^R d\varphi \, \bar{J}$. It is clear that the contribution of $\psi_2$ to the energy levels is the same as in the free theory. To find the contribution of $\psi_1$, we expand it in modes

$$\psi_1(\bar{z}) = \sqrt{\frac{1}{R'}} \sum_m \mathsf{b}_m \, e^{-\frac{2\pi i m \bar{z}'}{R'}} \,, \tag{3.33}$$

where $\mathsf{b}_m^\dagger = \mathsf{b}_{-m}$ and they obey the usual commutation relations

$$\{\mathsf{b}_m, \mathsf{b}_n\} = \delta_{m,-n} \,, \tag{3.34}$$

with $m$ an integer for Ramond boundary conditions and an integer plus a half for NS ones. The contribution of $\psi_1$ to the right-moving energy is then

$$E_R^{(1)} = \int_0^R d\varphi \, T_{\bar{z}\bar{z}}^{(1)} = \frac{i}{2} \int_0^R d\varphi \, \psi_1 \bar{\partial} \psi_1 \left(1 + \mu \bar{J}\right) = \frac{i}{2} \int_0^R d\varphi \, \psi_1 \bar{\partial}' \psi_1 \left(1 + \mu \bar{J}\right)^2 \,. \tag{3.35}$$

The two factors of $1 + \mu \bar{J}$ are cancelled by two corresponding factors of $R'^{-1} = (R - \mu Q)^{-1}$, one from the normalization of $\psi_1$ and one from the $\bar{z}'$ derivative. Therefore, we find that the energy spectrum is identical to that in the undeformed model, as expected.

# 4. Discussion and future directions

We have studied an irrelevant, yet integrable deformation of a general two-dimensional CFT possessing a chiral $U(1)$ current and worked out the finite-size spectrum and the thermodynamics of the resulting quasilocal QFT, in the special cases of a purely chiral/anti-chiral $U(1)$ current. In the chiral case, we found that the deformation acts non-trivially on the spectrum and may be induced via a field-dependent coordinate transformation on the original CFT that mixes left and right-movers. In the anti-chiral case, the spectrum is unchanged, and this can be understood from the fact that the deformation corresponds to a field-dependent, but purely antiholomorphic coordinate transformation of the original CFT, which is a symmetry. The discussion below will thus refer only to the non-trivial chiral case.

As in the $T\bar{T}$ case, the field-dependent coordinate transformation induces a change in the speed of sound, which can now be made superluminal for both signs of the deformation parameter $\mu$. While superluminality is not in itself a problem due to the lack of Lorentz invariance, note that this can lead to closed timelike curves in the finite-volume theory; in addition, we find that various thermodynamic quantities diverge if the circle on which the theory is placed

---

[12]To make this transformation more palatable, we could first bosonize $\psi_2$ to an antichiral boson, $H(\bar{z})$.

has radius $R < \mu Q/2$. Since both problems disappear as $R \to \infty$, we can still hope that the theory on the plane makes sense.

Given that the deformation involves following the RG flow upwards, understanding the UV behaviour of the deformed theory is non-trivial. In the $T\bar{T}$ case, much progress has been made using the $S$-matrix approach; more precisely, it was shown that $S$-matrix of the deformed theory only differs by an energy-dependent phase factor from the original one [27]. This phase factor modifies the high-enegy asymptotic behaviour of the S-matrix, preventing the UV completion from being a usual local QFT; however, it has been argued that such an asymptotic behaviour - termed asymptotic fragility - does not obviously lead to an inconsistency and should sometimes be allowed, e.g. in a theory of quantum gravity. It would be very interesting to investigate whether the effect of the $J\bar{T}$ deformation on the S-matrix can be similarly encompassed by a phase, and understand the type of asymptotic UV behaviour to which it leads.

Better understanding the UV behaviour of the deformed theory is also important from a technical point of view. In particular, our derivation of the deformed spectrum assumed that the spacetime and internal symmetries are associated to *local* conserved currents, i.e. that the deformed CFT can be approximated as a quasilocal QFT. This approximation is expected to break down at high enough energies, and we should gain a better understanding of its regime of validity. This limitation of our derivation exactly parallels that of [1] for the $T\bar{T}$ case, which is also expected to break down when the non-local nature of the deformed theory takes over. One important difference with respect to $T\bar{T}$ is that in our case the deformation parameter is a null vector, so one may expect that the non-localities are only restricted to the $x^-$ direction, while locality in $x^+$ is preserved. Note also that in the $T\bar{T}$ case, even though the derivation of [1] of the deformed spectrum may break down at scales of order $\mu$, the definition of the deformed theory via the S-matrix does appear to hold up to arbitrarily high energies, which is another reason that it would be worth finding such an alternative definition in the $J\bar{T}$ case.

Another technical point that would deserve a proper proof is to show that the deformed theory has exact $SL(2,\mathbb{R}) \times U(1)$ invariance, which is equivalent to showing that the deforming operator is exactly marginal with respect to the left conformal symmetries to all orders in conformal perturbation theory. We leave such a proof to later work.

As we already mentioned, one reason that $J\bar{T}$-deformed CFTs are interesting is that they may provide a first example of an $SL(2,\mathbb{R}) \times U(1)$-invariant QFT where the $U(1)$ is enhanced to a right-moving Virasoro symmetry. This expectation relies on our ability to treat the deformed CFTs as local two-dimensional QFTs (see discussion above), such that the results of [12] apply. That the right-moving translation symmetry should be enhanced to a full Virasoro (which in particular includes rescalings) sounds extremely counter-intuitive. In order to bring some support for this claim, in the appendix we work out a simple example involving a deformed classical free boson. We show that, indeed, the stress tensor can be made purely antiholomorphic on-shell, which is consistent with full Virasoro enhancement in the quantum theory. It would be extremely interesting to find this additional Virasoro in conformal perturbation theory.

It is worth emphasizing that, even if we find the above Virasoro symmetry, the underlying field theory is *not* a usual two-dimensional CFT, as can be concluded from its behaviour in finite volume. However, due to the potential non-localities discussed above, it remains to be seen to what extent this Virasoro symmetry exists and how it acts on the space of states.

Some other physical and technical points that would be worth understanding are: first, the effect of (chiral) anomalies on the spectrum and other physical properties of the theory; how to derive the spectrum in the case of more general $U(1)$ current, which is neither chiral nor anti-chiral; to consider more general, currents e.g. non-abelian ones. Also, it would be interesting to better understand the relationship between the theory on the cylinder and that

on the plane and how $J\bar{T}$ - deformed CFTs relate to warped CFTs.

Finally, let us end with some speculations on the possible implications of the $J\bar{T}$ deformation for holography. Two-dimensional holographic QFTs with $SL(2,\mathbb{R}) \times U(1)$ invariance and an infinitely-enhanced symmetry group have been extensively discussed in connection with the Kerr/CFT correspondence [28], a proposed microscopic description of maximally spinning black holes in our galaxy. The near-horizon region of the Kerr black hole is captured by a particular deformation of AdS$_3$ known as "warped AdS$_3$". A holographic study of this space-time suggests that its holographically dual QFT is a deformation of a two-dimensional CFT by an irrelevant $(1,2)$ operator [29, 30] and, just like in the case of the $T\bar{T}$ - type deformations discussed by [1], one is supposed to "go up" the RG flow. It is currently not understood what singles out the particular trajectory "up the RG flow" relevant for the Kerr/CFT case beyond the leading order in the deformation parameter $\mu$, though one piece of information is that it preserves the Cardy form of the thermal entropy. Also, the analysis of the asymptotic symmetries of warped AdS$_3$ spacetimes show an enhancement of the $U(1)$ right-moving translation symmetry to a full Virasoro symmetry.

The irrelevant operator used in defining the Kerr/CFT-type deformation is *not $J\bar{T}$*, since it needs to be a single-trace operator[13]. However, the two deformations do appear to have many features in common, such as the flow up the RG direction, the Cardy form of the entropy, the likely Virasoro enhancement of right-moving translations and the appearance of closed timelike curves in the finite volume theory [31]. Given that the $J\bar{T}$ deformation is defined also for finite $\mu$, we may hope to learn important lessons about Kerr/CFT by studying this much simpler theory. For example, as in $J\bar{T}$, the Cardy formula may simply follow from the fact that the spectrum is continuously deformed as a function of $\mu$; as for the Virasoro symmetry, we hope to obtain a concrete handle on establishing its existence, the degree to which it is a true symmetry, and compare with its holographic realisation.

# Acknowledgements

The author is grateful to C. Aron, J. Maldacena, K. Papadodimas, S. Sethi, N. Warner, X. Yin for interesting conversations and especially to A. Strominger and K. Zarembo for insightful conversations, encouragement and comments on the draft. Her work was supported by the ERC Starting Grant 679278 Emergent-BH, the Knut and Alice Wallenberg Foundation under grant 113410212 (as Wallenberg Academy Fellow) and the Swedish Research Council grant number 113410213.

# A.   Notes on the deformed free boson

In this appendix, we study the $J\bar{T}$ deformation of a two-dimensional free scalar, $X$, where $J$ is the current associated to shifts in $X$. This model is more complicated than the fermionic ones we studied in the main text - in particular, $J$ does not have definite chirality - and for now we do not have a solution for its spectrum. However, we find it worth pointing out a few facts, such as: i) the model is again related via a field-dependent coordinate transformation to the original free scalar; ii) at least for a simple class of solutions, parametrized by momentum and winding, we can explicitly check that the deformed energy spectrum obeys (2.16); iii) the right-moving stress tensor can always be improved (via a local on-shell redefinition) such that it becomes purely antiholomorphic.

---

[13]Explicit examples of which have been given in [30] for the specific case of deforming the D1-D5 CFT.

The last property applies to all the models described by the action (A.1) and brings preliminary evidence, so far at a purely classical level, that the right-moving translations are enhanced to a full right-moving Virasoro symmetry, in agreement with the results of [12].

We start from the action

$$S = \int d^2z \, \partial X \bar\partial X \, \mathcal{F}(\lambda \bar\partial X) \, , \tag{A.1}$$

where $\mathcal{F}$ is some arbitrary function with $\mathcal{F}(0) = 1$ and which admits a Taylor expansion around zero. By construction, this theory has $SL(2,\mathbb{R})_L \times U(1)_R$ invariance. The stress tensor is given by

$$T_{zz} = \frac{1}{2}(\partial X)^2 \big(\mathcal{F} + \lambda(\bar\partial X)\mathcal{F}'\big) \, , \qquad T_{\bar z z} = 0 \, , \tag{A.2}$$

$$T_{z\bar z} = \frac{\lambda}{2}\partial X(\bar\partial X)^2\mathcal{F}' \, , \qquad T_{\bar z\bar z} = \frac{1}{2}(\bar\partial X)^2\mathcal{F} \, . \tag{A.3}$$

As expected from the $SL(2,\mathbb{R})_L$ invariance, the component $T_{\bar z z} = 0$. The equation of motion is

$$\partial \bar\partial X \, (\mathcal{F} + \lambda \, \bar\partial X \, \mathcal{F}') + \lambda \, \partial X \, \bar\partial^2 X \, \mathcal{F}' + \frac{1}{2}\lambda^2 \partial X \, \bar\partial X \, \bar\partial^2 X \, \mathcal{F}'' = 0 \, , \tag{A.4}$$

and is not hard to check that $T_{zz}$ is holomorphic on-shell.

**The $J\bar T$ deformation**

Let us now consider the $U(1)$ current associated to the shift symmetry $X \to X + const$

$$J_z = \frac{1}{2}\partial X(\mathcal{F} + \lambda\bar\partial X\mathcal{F}') \, , \qquad J_{\bar z} = \frac{1}{2}\bar\partial X\mathcal{F} \, . \tag{A.5}$$

If we would like the theory (A.1) to correspond to the $J\bar T$ deformation, then we need

$$\partial_\lambda \mathcal{L} = J_z T_{\bar z\bar z} - J_{\bar z}T_{z\bar z} \, , \tag{A.6}$$

which should be true for any $\lambda$. Plugging in the expressions for $J_a$ and $T_{ab}$, we find that $\mathcal{F}' = \frac{1}{4}\mathcal{F}^2$, which allows us to solve for $\mathcal{F}$

$$\mathcal{F}(x) = \frac{1}{1 - x/4} \, . \tag{A.7}$$

Note that this form of the Lagrangian can be obtained from the free boson CFT by performing the field-dependent coordinate transformation

$$z \to z' = z \, , \quad \bar z \to \bar z' = \bar z - \frac{\lambda}{4}X \, . \tag{A.8}$$

For this particular choice of $\mathcal{F}$, the equation of motion simplifies to

$$\partial \bar\partial X + \frac{\lambda}{4}\partial X\bar\partial^2X\mathcal{F} = 0 \, . \tag{A.9}$$

To find the classical solutions, we note the above can also be written as $\bar\partial(\partial X\mathcal{F}) = 0$, from which we find that

$$\frac{\partial X}{1 - \frac{\lambda}{4}\bar\partial X} = holomorphic \, . \tag{A.10}$$

A general solution is

$$X(z,\bar z) = f(z) + g\big(\bar z - \frac{\lambda}{4}f(z)\big) \, , \tag{A.11}$$

where $f, g$ are arbitrary functions. Note that the deformation (A.1) with $\mathcal{F}$ given by (A.7) can also be interpreted as a *chiral $J\bar T$* deformation with $J = J_z = \mathcal{F}\partial X$.

**Basic spectrum check**

In principle, we could expand the above solution in modes (imposing the appropriate periodicity conditions), compute the conserved charges $Q$, $P$ and $E$ and check whether the $\mu$ dependence of the energy, at fixed $P$, $Q$, obeys (2.16). However, this appears tedious in practice, given the non-linearity of (A.11). We will therefore concentrate on a very simple solution

$$X(z,\bar{z}) = az + b\left(\bar{z} - \frac{\lambda}{4}az\right),\tag{A.12}$$

consisting of only momentum and winding[14], with no oscillators turned on. The constants $a$ and $b$ are expressed through the integer conserved charges $Q$ and $n = PR$ as

$$a = \frac{4QR - \epsilon\sqrt{2Q(2R - \lambda Q)(4QR - \lambda n)}}{\lambda QR},\tag{A.13}$$

$$b = \frac{2\left(\epsilon\sqrt{2Q(2R - \lambda Q)(4QR - \lambda n)} + 2\lambda Q^2 - 4QR\right)}{\lambda^2 Q^2 - 2\lambda QR},\tag{A.14}$$

where $\epsilon = \mathrm{sgn}\, Q$. Despite appearances, both are regular as $\lambda \to 0$. The expression we obtain for the energy is

$$E(\lambda, R) = \frac{Q\left(\lambda^2 n + 32R^2\right) - 4R\left(\lambda n + 2\epsilon\sqrt{2Q(2R - \lambda Q)(4QR - \lambda n)}\right) - 8\lambda Q^2 R}{\lambda^2 QR},\tag{A.15}$$

which again is regular at $\lambda = 0$. We notice the presence of a square root singularity at $R = \lambda Q/2$ - to be contrasted to the pole we obtained in the chiral $J$ case. We can explicitly check that

$$\partial_\lambda E(\lambda, R) = R(J_z T_{\bar{z}\bar{z}} - J_{\bar{z}} T_{z\bar{z}}),\tag{A.16}$$

thus confirming the expectation from the main text.

**Improvement of the stress tensor**

As is well known, the Noether procedure for constructing the stress tensor as the conserved current associated to translations may not yield a tensor with all the desired symmetry properties. It is possible in certain cases to 'improve' the stress tensor, enhancing its symmetries while leaving the conservation equations untouched. We will be interested in whether the right-moving part of the stress tensor can be improved

$$T_{z\bar{z}} \to \tilde{T}_{z\bar{z}} = T_{z\bar{z}} - \partial A, \qquad T_{\bar{z}\bar{z}} \to \tilde{T}_{\bar{z}\bar{z}} = T_{\bar{z}\bar{z}} + \bar{\partial} A,\tag{A.17}$$

such that $\tilde{T}_{z\bar{z}} = 0$. In that case, $\tilde{T}_{\bar{z}\bar{z}}$ will be entirely antiholomorphic on-shell.

The fact that the right-moving stress tensor can be improved such that it is antiholomorphic on-shell is true in all the theories of the form (A.1), not just for the $J\bar{T}$ deformation. To show this, we assume $\mathcal{F}(\lambda\bar{\partial}X)$ has a power series expansion of the form

$$\mathcal{F}(x) = 1 + ax + bx^2 + cx^3 + \dots,\tag{A.18}$$

and similarly for the classical solution

$$X(z,\bar{z}) = X^{(0)}(z,\bar{z}) + \lambda X^{(1)}(z,\bar{z}) + \lambda^2 X^{(2)}(z,\bar{z}) + \lambda^3 X^{(3)}(z,\bar{z}) + \dots\tag{A.19}$$

---

[14]In order to gain an extra parameter (winding), we are considering compact $X$.

The zeroth order solution is given by

$$X^{(0)}(z, \bar{z}) = x_L(z) + x_R(\bar{z}) \,, \tag{A.20}$$

and the higher order ones

$$X^{(1)}(z, \bar{z}) = -a\, x_L(z)\, x_R'(\bar{z}) \,, \tag{A.21}$$

$$X^{(2)}(z, \bar{z}) = \frac{1}{2}\left(a^2 x_L(z)^2 x_R''(\bar{z}) + 3a^2 x_L(z) x_R'(\bar{z})^2 - 3b\, x_L(z) x_R'(\bar{z})^2\right) \,, \tag{A.22}$$

etc, where we have set to zero the purely (anti)holomorphic solutions at higher order. The left-moving stress tensor, evaluated on this solution, is simply

$$T_{zz} = \frac{1}{2} x_L'(z)^2 \,, \tag{A.23}$$

and is purely holomorphic, as expected. Next, we plug these formulae into the series expansion of $T_{z\bar{z}}$ and find that the deformation is easily integrable with respect to $z$, if we choose

$$A = a\lambda\, x_L(z) x_R'(\bar{z})^2 - \lambda^2 x_L(z)\, x_R'(\bar{z})\left[\left(a^2 - 2b\right) x_R'(\bar{z})^2 + a^2 x_L(z) x_R''(\bar{z})\right] + \dots \tag{A.24}$$

The new antiholomorphic stress tensor is

$$\tilde{T}_{\bar{z}\bar{z}} = \frac{1}{2} x_R'(\bar{z})^2 \mathcal{F}(\lambda x_R'(\bar{z})) \,. \tag{A.25}$$

This confirms, at the classical level, the prediction of [12] that it is possible to redefine the stress tensor current so as to have a full RM Virasoro symmetry. Thus, the Virasoro symmetry does appear to exist explicitly. However, a curious feature of this redefinition is that while the energy is unchanged, the fact that $\tilde{T}_{z\bar{z}} = 0$ implies that $\tilde{T}_{xx} = -\tilde{T}_{\tau\tau}$, or $\partial_R E = -E/R$, which the deformed spectrum is unlikely to satisfy.

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
