# Peer review of "An integrable Lorentz-breaking deformation of two-dimensional CFTs"

_SciPost Physics, doi:SciPost Phys. 5, 048 (2018)_

## Round 1 · Referee Report · Edgar Shaghoulian · 2018-4-2

Strengths

1) An old, unsolved, important problem is being addressed. $\\$
2) A new tool is being brought to bear on the problem. $\\$
3) The results obtained are interesting and do not contradict the proposal. $\\$
4) The manuscript is well written.

Weaknesses

1) The calculations are almost identical to previous ones in the literature. $\\$
2) The meaning of the proposed deformation is not discussed clearly, propagating confusion from previous literature.

Report

This paper provides concrete calculations for a solvable irrelevant deformation of conformal field theory. The calculations are correct and are supplemented by simple toy theories. The problem is interesting purely from the point of view of trying to construct exotic field theories, but the real force of the paper is to gain insight into a precise description of near-extremal black holes, which realize a symmetry structure like that of the field theories being analyzed. There are some issues with the conceptual grounding of the calculations which I will address in the “Requested changes” section.

Requested changes

1) Expressions like 1.1, 2.1, 2.5, and 2.17 are mnemonics which are incorrect if interpreted literally. The Smirnov-Zamolodchikov deformations are defined as
\[
\frac{dS(\mu)}{d\mu} = \int d^2 z \left(T_s \bar{T}_{s'}-\Theta_{s-2}\bar{\Theta}_{s'-2}\right)_\mu
\]
which, importantly, is different than what is written in the paper. The reason these are different is because the operator that appears on the RHS of what I have written above is a function of $\mu$, indicated by the subscript. Specializing to the $J\bar{T}$ deformation considered in this paper, the deformation is \emph{not} the same as adding the CFT operator $J\bar{T}$ as an irrelevant deformation to the action. (It only agrees with this at first order in $\mu$, but the calculations being done in the paper are nonperturbative in $\mu$.) Nor is it the same as adding the operator $(J\bar{T})_\mu$ to the action, since then differentiating with respect to $\mu$ would not give the equation above. $\\$
2) The OPE 2.3 needs clarification. This is correct for the operator $J\bar{T}$ with $J$ and $\bar{T}$ the CFT operators. But as discussed in point (1) the RG flow is defined by the flowed operators via the equation written in (1). Is it an assumption that this operator continues to be well-defined along the flow? If so this assumption should be stated explicitly. In particular the UV is no longer an ordinary CFT with a convergent OPE upon deformation. $\\$
3) Below equation (2.9), it seems unnecessary to take the limit $z \rightarrow z’$ to rewrite the correlator as $\langle n | \mathcal{O}_{J\bar{T}}|n\rangle$, you can simply insert the OPE. $\\$
4) Above equation (2.11), the only assumption needed here is that $|n\rangle$ is non-degenerate, not that the entire spectrum is non-degenerate. $\\$
5) At the end of the subsection “Purely holomorphic current” in section 2.1, it says “but rather a pole at a finite energy.” The wording is confusing; the pole is presumably referring to the one at finite $\mu$ (or $R$ or $Q$) which makes all energies infinite. $\\$
6) Footnote 7 needs clarification. I understand the formula for $\delta S$ in CFT, as long as (a) we are working to first order in $\xi$, (b) there is a $\sqrt{g}$ factor in the integral, and (c) without any total derivative terms. You mention a comparison to a different form of the variation with $\sqrt{g}$ in it, but I still do not understand this equation. $\\$
7) Below equation (2.42) the phrase “boundary metric” is used where I believe simply “metric” is meant. $\\$
8) In equation (2.45) it should be $\bar{J}(\bar{z}’}$ in the integrand. $\\$
9) In the third paragraph of section 4 a reference is missing. $\\$
10) In the final paragraph in section 4 you discuss that the Kerr/CFT-type deformation needs to be single trace. I think the contextualization would improve dramatically if you discussed why this is. In particular, the Smirnov-Zamolodchikov type double-trace deformations have an effect on classical gravity in the bulk, since e.g. the case of $T\bar{T}$ corresponds to a finite radial cutoff. So naively even though they are double trace they can affect important leading-in-$N$ quantities. Furthermore, an obvious thing to consider, which would be nice to comment on if you have, is a Giveon-Itzhaki-Kutasov style single-trace $J\bar{T}$ deformation, i.e. consider the symmetric orbifold of a $J\bar{T}$ deformed CFT.

---

## Round 1 · Referee Report · Anonymous · 2018-4-30

Strengths

1- Gives compelling motivation for the study of $J\bar{T}$ deformation of CFTs.
2- Derives exact results and checks them in an example using alternative methods.

Weaknesses

1- In many instances the $J\bar{T}$ deformation is treated as if it was just what is given in (2.1), instead
\[{dS\over d\mu}=\int d^2z\, {\cal O}_{J\bar{T}}(\mu)\]
is the action for which the methods of [1] apply.

Report

The paper "An integrable Lorentz-breaking deformation of two-dimensional CFTs" is a beautiful original study in the topic of integrable irrelevant deformations of 2d CFTs, which is a highly active field of research. The author determines the deformation of the spectrum of $J\bar{T}$ (for chiral $J$), and uses these results to draw conclusions about the UV-completeness of this theory, and speculate about the potential role it can play in holography. I very strongly recommend it for publication, after its weakness (given above) has been addressed.

Besides the requested change, I also recommend that the author explains why the highly unusual operator valued spacetime transformations do not lead to problems. I would also be interested in how to understand the agreement between the energy in classical field theory (A.15) and an energy egienvalue of the quantum Hamiltonian. (In many situations there is no such relation between the classical and quantum theory.)

Requested changes

1- Fix all occurences of the imprecisions resulting from the difference between ${dS\over d\mu}=\int d^2z\, {\cal O}_{J\bar{T}}(\mu)$ and $S=S_\text{CFT}+\mu \int d^2z\, {\cal O}_{J\bar{T}}(0)$.

A partial list is (2.1), (2.5), and the discussion after (2.39).

  • validity: high
  • significance: top
  • originality: high
  • clarity: high
  • formatting: perfect
  • grammar: perfect

Author:  Monica Guica  on 2018-07-17  [id 292]

(in reply to Report 2 on 2018-04-30)
Category:
answer to question
correction

First, I would like to thank the referees for their comments. I have implemented most of the required modifications as follows:

  • I have changed eqns (1.1), (2.1), (2.17) and removed eqn (2.5), to comply with both referees' requests for a more clear notation

  • I have added footnote 3 with comments about the validity of the OPE. This should address comment 2 of referee 1

  • I corrected the type-os that were pointed out in comments 7,8,9 of referee 1, and removed the paragraph referred to in comment 5

Additionally, I have added/updated a few references and included some comments about anomalies.

Here are also some answers to other issues raised by the referees:

One important issue that was raised was to explain why the highly unusual operator-valued spacetime transformations do not lead to problems. I agree with referee 2 that this is an important problem that needs to be addressed, as at the moment it is not even clear what is meant precisely by such a transformation (e.g., are there normal-ordering issues?). In the paper, whenever such a transformation was invoked, I was working in a superselection sector with <J> fixed, precisely in order to avoid such problems.

For comment 6) of referee 1 concerning clarifications to footnote 7: the main purpose of the footnote is to define the stress tensor that will be used in later sections, which happens to have a \sqrt{g} inside, and thus differs by a factor of 2 from the usual stress tensor. This should answer comment b). For comment a), the text already mentions this analysis is perturbative and for c), the symmetry variation of the action need not equal the Noether current, there can be total derivative terms.

As for comment 10) of referee 1, I would like to point the referee to a paper I wrote that is dedicated entirely to the holographic interpretation of JTbar-deformed CFTs (https://arxiv.org/abs/1803.09753) . There are also two recent papers that discuss the single-trace analogue of the JTbar deformation. The readers who are interested in the holographic applications should probably consult these more recent works.

---

## Round 2 · List of Changes

comments and references added, a factor of 2 corrected

You are currently on this page

Resubmission 1710.08415v2 on 18 September 2018

---

## Editorial Decision

published